# Metabolic Features of Saliva in Breast Cancer Patients

**DOI:** 10.3390/metabo12020166

**Published:** 2022-02-10

**Authors:** Lyudmila V. Bel’skaya, Elena A. Sarf, Denis V. Solomatin, Victor K. Kosenok

**Affiliations:** 1Biochemistry Research Laboratory, Omsk State Pedagogical University, 14, Tukhachevsky str., 644043 Omsk, Russia; nemcha@mail.ru; 2Department of Mathematics and Mathematics Teaching Methods, Omsk State Pedagogical University, 14, Tukhachevsky str., 644043 Omsk, Russia; denis_2001j@bk.ru; 3Department of Oncology, Omsk State Medical University, 12, Lenina str., 644099 Omsk, Russia; victorkosenok@gmail.com

**Keywords:** breast cancer, saliva, biochemistry, diagnosis

## Abstract

The aim of the work was to study the metabolic characteristics of saliva in breast cancer and the subsequent assessment of the potential information content of its individual biochemical indicators. The study included 487 patients of the Omsk Clinical Oncology Center with morphologically verified breast cancer and 298 volunteers without breast pathologies. Saliva samples were obtained from all patients before the start of treatment, and the values of 34 biochemical indicators were determined. It has been shown that concentration of total protein, urea, uric acid (UA), the total content of α-amino acids and lipid peroxidation products, and the activity of metabolic and antioxidant enzymes (in particular catalase—CAT) of saliva changed significantly in breast cancer. Biochemical indicators characterizing early breast cancer have been identified, which can be used for timely diagnosis in addition to existing methods. The coefficients UA/Urea and UA·CAT/Urea are proposed, for which the maximum deviation from the norm was observed in the early stages of the disease. It was shown that for ductal breast cancer, changes in the activity of metabolic enzymes of saliva were more pronounced, while, for lobular breast cancer, the indicators of enzymatic and non-enzymatic components of antioxidant protection changed. The results confirmed the potential importance of saliva in the diagnosis of breast cancer.

## 1. Introduction

Breast cancer (BC) is the most common female cancer worldwide. According to GLOBOCAN 2020, breast cancer is the leader in the structure of global incidence (24.5% of all malignant tumors) and consistently ranks first in the structure of global mortality in women (15.5% of deaths from malignant tumors) [1]. The incidence rates of breast cancer in the world over the past 30 years have increased, which is associated with an absolute and relative increase in incidence [2]. However, the 5-year survival rate for breast cancer can approach 90% in developed countries when it is detected at an early stage [3]. Thus, early detection of breast cancer is important to improve prognosis and survival [4,5].

Technologies that allow for detecting breast cancer at an early stage include digital mammography, magnetic resonance imaging of the mammary glands with contrast, elastography [6]. Plasma carcinoembryonic antigen (CEA) and cancer antigen 15–3 (CA 15–3) are the most commonly used tumor markers in breast cancer [7,8,9]. Tumor markers are considered a non-invasive and cost-effective way to monitor the course of the disease, determine the prognosis and plan treatment by predicting the response to treatment [10], but their applicability for the diagnosis of breast cancer has not been confirmed [11,12].

In the last decade, interest has increased in the use of saliva as an additional test that improves traditional approaches to the medical evaluation of serious systemic diseases, including the detection and screening of breast cancer [13,14,15,16]. Potentially informative salivary tumor markers in breast cancer are epidermal growth factor (EGF), human epidermal growth factor receptor 2 (HER2), vascular endothelial growth factor (VEGF), carcinoembryonic antigen (CEA), CA15-3, and oncogenic tumor suppressor protein (p53) [17,18,19,20]. Recent “omics” technologies, such as transcriptomics, proteomics, and glycoproteomics, due to their ability to simultaneously quantify hundreds of molecules and patterns, have been used to distinguish breast cancer patients from healthy people [13,21,22,23,24]. The use of salivary volatile organic compounds for the diagnosis of breast cancer has been described [25]. It has been reported that hydrophilic metabolites, such as amino acids and polyamines, can be used to differentiate breast cancer patients from healthy individuals [26,27,28,29]. In general, the dynamics of individual components of saliva in breast cancer (trace elements, amino acids, fatty acids, cytokines, etc.) has been widely studied [30,31,32,33]. However, an integrated approach to the study of saliva for its use in the diagnosis of breast cancer has not yet been implemented.

Saliva is an alternative to blood and other biological fluids as a source of potential biomarkers [16]. The main advantage of saliva is its easy non-invasive collection. Studies devoted to the analysis of the proteome and transcriptome of saliva have shown the great potential of this biological fluid for the detection of breast cancer [13,34]. The existence of a positive correlation between the expression of a number of biomarkers in blood serum and in saliva has been previously shown, since patients with cancer have a higher expression of specific biomarkers compared to patients without cancer [17,35,36,37]. However, most of the studies included patients in the advanced stages of the disease, when the patient’s prognosis was poor, and other diagnostic methods are more effective for making a diagnosis [13]. The identification of cancer at an early stage is extremely important to ensure a good prognosis for the patient and to provide therapy that reduces disease progression and death [38]. 

The concept of the study stems from the fact that all biomarkers of breast cancer that are technically possible have been identified; there is no universal diagnostic indicator or a combination of complementary indicators of breast cancer, which is due to the significant heterogeneity of the disease. The search for a combination of potential biomarkers is most promising in saliva, which we have shown earlier using lung cancer as an example. It was especially important to analyze the possibility of detecting early breast cancer using saliva [39].

In the present study, 34 biochemical indicators of saliva in breast cancer were identified and their potential significance was assessed. This study was one of the largest and included patients with early stages of breast cancer (226/487). The metabolic features of the composition of the saliva of patients depending on the prevalence of the process and the histological type of breast cancer were considered. Biochemical indicators of saliva were identified, the change of which is typical for the early stages of breast cancer.

## 2. Results

### 2.1. Metabolic Features of Saliva Composition in Breast Cancer 

We observed a change in the metabolic profile of saliva in breast cancer (Table 1). 

Thus, with breast cancer, the content of total protein in saliva decreased (−40.7%) with a simultaneous increase in the level of α-amino acids (+4.2%). An increase in the activities of metabolic enzymes LDH (+31.7%), ALP (+19.7%), GGT (+13.7%) and α-Amylase (+65.5%) was observed. The proportion of toxic lipid peroxidation products SB/TC-ratio (+3.0%) and MDA (+9.1%) increased against the background of a decrease in catalase activity (−17.5%) and uric acid content (−20.5%).

Analysis of the entire group of patients with breast cancer compared with healthy controls by principal component analysis (PCA) showed that the groups differed significantly (*p* = 0.0014, Figure 1a). 

The contribution of biochemical indicators of saliva to the division of groups was ambiguous: for PC1, medium-strength correlations were established for protein (*r =* 0.6680), catalase (*r* = 0.6420), LDH (*r =* 0.5942), ALP (*r* = 0.5616), GGT (*r =* 0.5395), α-amino acids (*r =* 0.5086), urea (*r =* 0.4474) and α-amylase (*r =* 0.4083). For PC2, only two strong correlations were noted with the content of Schiff bases *(r* = 0.7515) and the SB/TC-ratio (*r* = 0.7784, *p* < 0.0001). The use of the listed biochemical indicators for the construction of the random forest method classifier made it possible to obtain sensitivity values of 84.53%, specificity of 75.00%. When assessing the quality of classification by the AUC-ROC metric, it was shown that the area under the curve was 0.9091 (Figure 1b). We have shown the fundamental possibility of constructing a diagnostic scheme based on a decision tree using nine biochemical salivary indicators (Figure 1c). When constructing classification trees, we took into account the division into two groups (BC/Control). Sensitivity was shown to be as high as 91.72%, while specificity was only 60.40%.

### 2.2. Metabolic Features of Saliva Composition Depending on the Stage of Breast Cancer

At the next stage, we compared the biochemical composition of saliva depending on the prevalence of breast cancer (Table 2). It was shown that the same biochemical indicators change statistically significantly in comparison with healthy controls as, in general, for breast cancer. Additionally, seromucoids and Ca/P-ratio were included in the list of indicators, Na/K-ratio and SOD/Catalase-ratio were excluded (Table 1 vs. Table 2). In the early stages of breast cancer (Groups I, and II), there was a minimum content of total protein and uric acid, a maximum content of urea and catalase activity (Table 2).

Since the content of urea, uric acid (UA) and catalase activity (CAT) vary in different directions, we proposed the coefficients “UA/Urea” and “UA∙CAT/Urea” (Table 3). These coefficients made it possible to distinguish between patients with breast cancer and healthy ones; the maximum differences were noted for early breast cancer (Group I).

Multivariate assessment of biochemical indicators of saliva by PCA showed that PC1 separated healthy controls and patients with breast cancer (*p* < 0.0001), while PC2 separated early and advanced stages of breast cancer (Figure 2A). A clear division was observed between groups 1 and 4, while groups 2 and 3 actually coincided (Figure 2A). The division according to the first main component correlated with the content of seromucoids (*r* = 0.6694), total protein (*r* = 0.6566), α-amino acids (*r* = 0.5747), urea (*r* = 0.5011), LDH activity (*r* = 0.5429), ALP (*r* = 0.5408), GGT (*r* = 0.5368), α-amylase (*r* = 0.4871) (Figure 2B). For PC2, there was a strong correlation with the SB/TC-ratio (*r* = 0.7591) and medium-strength correlations with the content of uric acid (*r* = 0.4536), urea (*r* = −0.4397) and α-amino acids (*r* = −0.4942) (Figure 2B). The separation of the control group and groups 1 and 4 was statistically significant (*p* < 0.0001).

### 2.3. Metabolic Features of Saliva Composition Depending on the Histological Type of Breast Cancer 

At the first stage, PCA analysis showed that ductal and lobular breast cancer differ from each other (Figure 3). Thus, PC1 allowed to separate the group with ductal breast cancer (to the right of the vertical axis, *p* = 0.0140), while the groups with lobular breast cancer and the control group represented a single field in the diagram (Figure 3A). Strong correlations were found for MM 254 (*r* = 0.7818) and MM 280 (*r* = 0.7742), medium-strength correlations were found for total protein (*r* = 0.6673), catalase (*r* = 0.5567), α- amino acids (*r* = 0.5479), LDH (*r* = 0.5465), ALP (*r* = 0.5059), GGT (*r* = 0.4790), urea (*r* = 0.4377) and α-amylase (*r* = 0.4271).

With different histological types of breast cancer, different biochemical indicators of saliva changed (Table 4). Thus, for lobular breast cancer, the maximum decrease in the level of protein toxins (MM 254 and MM 280) was noted with a simultaneous increase in the content of lipid peroxidation products (TC and SB). In addition, in lobular breast cancer, a decrease in catalase activity and uric acid concentration was more pronounced, while, in ductal breast cancer, the changes concerned AOA and salivary peroxidase activity (Table 4). It should be noted that, for ductal breast cancer, changes in the activity of metabolic enzymes were more pronounced than in healthy controls. Among themselves, groups with ductal and lobular breast cancer differed in the content of medium molecular weight toxins and GGT activity (Table 4).

## 3. Discussion

We have shown that in breast cancer there is a decrease in the content of total protein, which is consistent with the literature data [22]. The reason for the decrease in total protein in saliva may be related to the transport system, in which transport depends on the polarity and charge of the molecule. So far, most of the biomarkers identified are not intrinsic to saliva but are serum-derived low molecular weight inflammatory markers that are carried into saliva. In another study on the example of 20 patients with breast cancer and 11 healthy volunteers, patients with breast cancer showed a significant decrease in the level of total protein, pH, and lipid peroxidation products [40]. We have not shown any significant changes in pH, while the content of lipid peroxidation products increased. An increase in the level of lipid peroxidation products was also accompanied by a shift in the equilibrium towards more toxic Schiff bases, which was characterized by an increase in the SB/TC-ratio. At the same time, the activity of antioxidant enzymes (catalase) and the content of uric acid, as the main component of the non-enzymatic link in the antioxidant defense system of saliva, decreased [41]. We have shown similar changes earlier in the example of lung cancer, which suggests a similar pattern of changes in the lipid peroxidation–antioxidant defense system of saliva in oncological diseases [39]. Gornitsky et al. showed significantly higher concentrations of a marker of oxidative damage to DNA (8-oxo-7-hydrodeoxyguanosine) in saliva [42]. It is known that oxidative stress is the main phenotypic feature in patients with breast cancer [43,44]. However, very little is known so far about the use of salivary oxidative stress biomarkers to diagnose breast cancer [45]. Oxidative stress can be detected by assessing the rate of synthesis of reactive oxygen species, the concentration/activity of enzymatic and non-enzymatic antioxidants, as well as the level of products of oxidative modification of proteins and lipids. Disturbances in the mechanisms of antioxidant defense (a decrease in the concentration of reduced glutathione, an increase in superoxide dismutase activity, a decrease in the activity of glutathione peroxidase) and the intensity of oxidative damage to cells (an increase in the content of MDA, protein carbonyl groups and 8-OHdG) were previously shown in the blood serum of patients with BRCA1-dependent breast cancer [46,47].

In a study [45], catalase activity did not differ between groups, while salivary peroxidase activity in saliva was significantly higher in patients with BRCA1-dependent breast cancer compared with the control group. Salivary peroxidase performs a dual role: it is responsible for the breakdown of cytotoxic hydrogen peroxide and has bactericidal activity against *Streptococcus mutans*, *Escherichia coli*, *Salmonella typhimurium*, and various species of *Lactobacillus* and *Actinomyces* [48]. It should be emphasized that salivary peroxidase is the only antioxidant synthesized exclusively in the salivary glands [29,31]. Thus, salivary peroxidase activity reflects the prevention of oxidative stress by the salivary glands. An analysis of the data obtained by the authors [45] suggested that an increase in peroxidase activity in the saliva of patients with BRCA1-dependent breast cancer indicated an increase in the enzymatic antioxidant defense that protects the salivary glands and the entire oral cavity from oxidative damage. Nevertheless, our study showed a decrease in catalase activity in all groups of patients with breast cancer, reaching a minimum value in metastatic breast cancer (Table 2). At the same time, an increase in SOD activity was observed, which was consistent with the previously obtained data. A statistically significant increase in peroxidase activity was noted only for ductal breast cancer (Table 4).

Breast cancer patients showed higher concentrations of polyamines and amino acids in saliva than controls [24]. The increased concentration of amino acids in saliva was consistent with another report [29]. We have shown an increase in the total content of α-amino acids in saliva both in general for the group of breast cancer patients and depending on the stage of the disease (Table 1 and Table 2). It was interesting to note that the maximum increase was observed in the early stages for ductal breast cancer (Table 4). An increase in the content of free amino acids was observed against the background of a decrease in the content of total protein, which may indicate a pronounced increase in protein catabolism. An additional confirmation of this assumption was the increase in the content of saliva urea as the end product of protein metabolism in the body (Table 1, Table 2 and Table 3).

An increase in the activity of metabolic enzymes in saliva in breast cancer patients was an interesting fact (Table 2). An increase in the activity of LDH, the final enzyme of anaerobic glycolysis, was natural due to an increase in glycolysis along with a low efficiency of mitochondrial oxidation in breast cancer [49,50]. Several studies have shown that overexpression of ALP may be associated with metastatic processes in cancers, including breast cancer [51,52]. Both LDH and ALP showed high activity in the early stages, which slightly decreases at advanced stages of breast cancer (−15.2 and −5.8% for LDH and ALP, respectively) and increases sharply in metastatic cancer (+33.1 and + 15.2% for LDH and ALP, respectively). For GGT, an increase in activity was observed only in the early stages, and then the activity decreases (Table 2). Elevated GGT has been known to be an independent risk factor for endometrial cancer, especially in postmenopausal and obese premenopausal women [53]. In an experiment on mouse tumors, significantly increased GGT activity was found in transplanted spontaneous breast carcinoma compared to normal or lactating mammary glands [54]. Human breast tumors showed significantly elevated enzyme levels compared to normal tissue or histologically unaffected breast tissue containing carcinoma [55]. The α-amylase activity was significantly increased and reached its maximum value in metastatic breast cancer (Table 3). Previously, we had shown this for lung adenocarcinoma [56], which is consistent with the literature data [57]. For all enzymes, a statistically significant increase in activity was characteristic of ductal breast cancer (Table 4).

Our study has several limitations. The list of detectable salivary indicators was compiled based on the possibility of their determination using a semi-automatic biochemical analyzer. Thus, we planned to test the potential significance of their analysis in breast cancer and to understand the feasibility of a more in-depth study of individual indicators. Based on the results obtained, we further planned to determine the activity of LDH and ALP isoenzymes, the content of free amino acids, proteins and their molecular weight, etc. The limitations included the absence of patients with non-malignant pathologies of the mammary glands (fibroadenoma), which can be considered as a comparison group. In addition, at this stage of the study, we did not consider the change in the metabolic profile of saliva depending on the tumor phenotype.

## 4. Materials and Methods

### 4.1. Study Design and Group Description

The study included 487 patients of the Clinical Oncological Dispensary in Omsk. The sample size of this study was the number we could recruit within the study periods (January 2015–May 2017). All patients had been histologically diagnosed with breast cancer. None had received any prior treatment, including hormone therapy, chemotherapy, molecularly targeted therapy, radiotherapy, surgery, etc. The inclusion criteria were considered: the age of patients 30–70 years, the absence of any treatment at the time of the study, the absence of signs of active infection (including purulent processes), and good oral hygiene. Good oral hygiene meant clean teeth free of stuck food; pink gums that don’t hurt or bleed when brushing and no bad breath or pain. We took into account that patients did not have untreated dental caries and periodontal disease, which could affect the results of the study. The volunteers included in the study did not reveal any clinically significant concomitant diseases other than cancer pathology (in particular, diabetes mellitus, cardiovascular pathologies, etc.) that could affect the results of the study. Exclusion criteria: lack of histological verification of the diagnosis.

A detailed description of the study group is given in Table 5. Additionally, the study group was divided into the following subgroups: primary resectable breast cancer (Group I-T_1-2_N_0_M_0_, *n* = 226), treatment tactics were selected taking into account the biological subtype of breast cancer (Group II-T_1-3_N_1_M_0_ and T_3_N_0_M_0_, *n* = 131), primary inoperable breast cancer (Group III-T_0-3_N_2_M_0_, T_4_N_0-2_M_0_ and T_0-4_N_3_M_0_, *n* = 75) and disseminated breast cancer (Group IV-T_0-4_N_0-3_M_1_, *n* = 55). TNM classification has been made in accordance with AJCC (8th edition, 2017). The control group consisted of 298 healthy patients, in whom no breast pathology was detected during routine clinical examination.

The study was approved at a meeting of the Ethics Committee of the Omsk Regional Clinical Hospital “Clinical Oncology Centre” on 21 July 2016 (Protocol No. 15). All of the volunteers provided written informed consent.

### 4.2. Collection, Processing, Storage and Analysis of Saliva Samples

Saliva (5 mL) was collected from all participants prior to treatment. Collection of saliva samples was carried out on an empty stomach after rinsing the mouth with water in the interval of 8–10 am by spitting into sterile polypropylene tubes; the salivation rate (mL/min) was calculated. We did not find significant differences in the salivary flow rate in the studied groups, so they are not shown in the tables below. Saliva samples were centrifuged (10,000× *g* for 10 min) (CLb-16, Moscow, Russia), after which biochemical analysis was immediately performed without storage and freezing using the StatFax 3300 semi-automatic biochemical analyzer (Awareness Technology, Palm City, FL, USA) [39].

The pH, mineral composition (calcium, phosphorus, sodium, potassium, magnesium, chlorides), the content of urea, total protein, albumin, uric acid, α-amino acids, imidazole compounds, seromucoids and sialic acids and the activity of enzymes (aminotransferases—ALT, AST; alkaline phosphatase—ALP; lactate dehydrogenase—LDH; gamma-glutamyl transpeptidase—GGT; α-amylase) were determined in all samples. The content of substrates for lipid peroxidation processes (diene conjugates—DC, triene conjugates—TC, Schiff bases—SB, malondialdehyde—MDA) and the level of medium molecular weight toxins (MM) were determined. We determined the MM at wavelengths of 254 and 280 nm; they are designated MM 254 and MM 280, respectively [58]. Additionally, we assessed the activity of antioxidant enzymes (catalase, superoxide dismutase—SOD) and antioxidant activity. The potential value of calculating ratios such as “Na/K-ratio”, “Ca/P-ratio”, “SOD/Catalase-ratio” and “SB/TC-ratio” has been preliminary shown. In the article, tables show only indicators for which differences in values between groups are statistically significant. A complete list of the values of the determined indicators is given in Appendix A.

### 4.3. Statistical Analysis

Statistical analysis was performed using Statistica 13.3 EN software (StatSoft, Tulsa, OK, USA); R version 3.6.3; RStudio Version 1.2.5033; FactoMineR version 2.3. (RStudio, version 3.2.3, Boston, MA, USA) by a nonparametric method using the Mann–Whitney U-test and the Kruskal–Wallis H-test. The description of the sample was made by calculating the median (Me) and the interquartile range as the 25th and 75th percentiles (LQ; UQ). Differences were considered statistically significant at *p* < 0.05. 

A principal component analysis (PCA) was performed using the PCA program in R [59]. The choice of variables for the PCA method was carried out according to the results of comparison of biochemical indicators in the studied groups. When comparing two groups, we used the Mann–Whitney test; when comparing three groups or more, we used the Kruskal–Wallis test. Next, we selected indicators for which the differences between all groups are significant at the *p* < 0.10 level. PCA results are presented in the form of factor planes and corresponding correlation circles. In each case, the figures show only the first two principal components (PC1 and PC2). The color of the arrows on the correlation circle changed from blue (weak correlation) to red (strong correlation), as shown on the color bar. The orientation of the arrows characterized positive and negative correlations (for the first principal component, we analyzed the location of the arrows relative to the vertical axis; for the second principal component, relative to the horizontal axis). The significance of the correlation was determined by the correlation coefficient (r): strong—*r* = ±0.700 to ±1.00, medium—*r* = ±0.300 to ±0.699, weak—*r* = 0.00 to ±0.299. 

To construct classification trees, the exhaustive search method for one-dimensional branches CART (classification and regression tree) was used. In the diagrams, ID is the number of the vertex, N is the number of objects directed along this branch, branching conditions are indicated near each vertex, and the diagram inside each vertex shows the classification result; if all the observations are classified correctly, then the column corresponding to the predicted class will be high, and the rest are small.

## 5. Conclusions

Certain metabolic features characterized the composition of saliva in breast cancer. In accordance with the concept of the study, nine biochemical salivary indicators were identified to differentiate patients with breast cancer from healthy controls with a sensitivity of 91.72%. These indicators were total protein, urea, α-amino acids, nitric oxide, MDA, GGT, alkaline phosphatase, as well as SB/TC-ratio and Na/K-ratio. Biochemical indicators were identified, the change in which characterizes early breast cancer, which can be used for timely diagnosis in addition to existing methods. For these purposes, the ratios UA/Urea and UA∙CAT/Urea were calculated. It was shown that, for ductal breast cancer, the activities of metabolic enzymes of saliva changed greatly, while, for lobular breast cancer, changes in the indicators of enzymatic and non-enzymatic components of antioxidant protection were more pronounced. The obtained data emphasize the expediency of further study of the metabolic characteristics of saliva in relation to the diagnosis of breast cancer.

## Figures and Tables

**Figure 1 metabolites-12-00166-f001:**
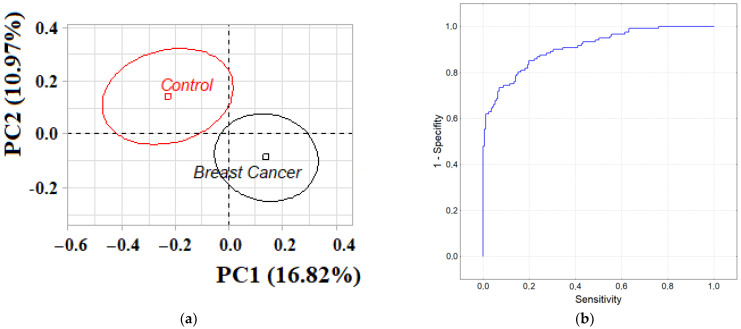
(**a**) PCA Factor plane for two groups (BC and Control); (**b**)—ROC curve for random forest method; (**c**)—Tree graph. Num. of non-terminal nodes—10, num. of terminal nodes—11. Model: C&RT. ALP—alkali phosphatase, MDA—malondialdehyde, GGT—gamma glutamyltransferase, SB—Schiff Bases, TC—triene conjugates.

**Figure 2 metabolites-12-00166-f002:**
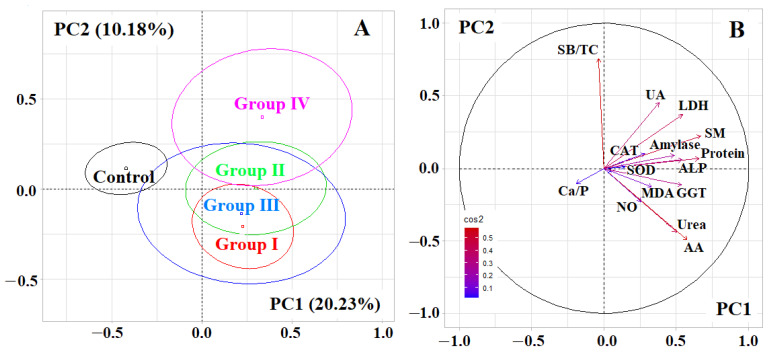
(**A**)—Individuals factor map (PCA); (**B**)—Variables factor map. UA—uric acid, LDH—lactate dehydrogenase, SM—seromucoids, CAT—catalase, ALP—alkali phosphatase, MDA—malondialdehyde, GGT—gamma glutamyltransferase, AA—α-Aminoacids, SB—Schiff Bases, TC—triene conjugates, SOD—superoxide dismutase.

**Figure 3 metabolites-12-00166-f003:**
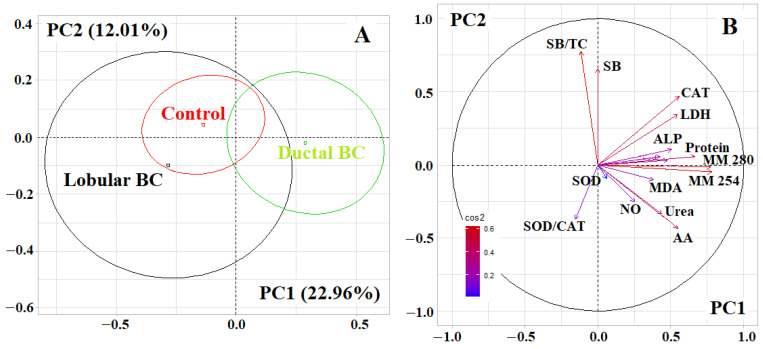
(**A**)—Individuals factor map (PCA); (**B**)—Variables factor map. UA—uric acid, LDH—lactate dehydrogenase, CAT—catalase, ALP—alkali phosphatase, MDA—malondialdehyde, GGT—gamma glutamyltransferase, AA—α-Aminoacids, SB—Schiff Bases, TC—triene conjugates, SOD—superoxide dismutase, MM—middle molecules.

**Table 1 metabolites-12-00166-t001:** Biochemical composition of saliva in breast cancer patients and healthy controls.

Indicator	Breast Cancer, *n* = 487	Control Group, *n* = 298	*p*-Value
Protein, g/L	0.64 [0.37; 1.09]	1.08 [0.65; 1.70]	0.0000
Urea, mmol/L	9.63 [6.25; 13.38]	6.66 [4.36; 9.13]	0.0000
Uric acid, μmol/L	65.4 [24.1; 136.1]	85.9 [34.4; 144.5]	0.0119
α-Aminoacids, mmol/L	4.23 [3.89; 4.76]	4.06 [3.83; 4.32]	0.0000
NO, μmol/L	29.1 [17.4; 44.6]	22.8 [13.2; 36.8]	0.0001
ALP, U/L	72.8 [47.8; 106.5]	60.8 [41.3; 84.7]	0.0002
LDH, U/L	1451.0 [861.6; 2093.0]	1101.5 [635.7; 1908.0]	0.0002
Catalase, nkat/mL	3.78 [2.53; 5.99]	4.58 [3.32; 5.79]	0.0052
MDA, μmol/L	7.09 [5.81; 8.97]	6.50 [5.73; 7.95]	0.0006
GGT, U/L	23.2 [20.0; 26.5]	20.4 [17.4; 24.4]	0.0000
SOD, c.u.	73.7 [34.2; 142.1]	57.9 [31.6; 113.2]	0.0247
α-Amylase, U/L	306.5 [122.6; 605.3]	185.2 [83.5; 384.4]	0.0002
Na/K-ratio, c.u.	0.686 [0.481; 1.067]	0.751 [0.505; 1.190]	0.1270
SOD/Catalase-ratio, c.u.	19.8 [8.4; 39.7]	14.5 [6.6; 37.7]	0.0168
SB/TC-ratio, c.u.	0.627 [0.561; 0.704]	0.609 [0.567; 0.667]	0.0503

Note. NO—nitric oxide, ALP—alkali phosphatase, LDH—lactate dehydrogenase, MDA—malondialdehyde, GGT—gamma-glutamyl transpeptidase, SOD—superoxide dismutase, SB—Schiff bases, TC—triene conjugates, c.u.—conventional units.

**Table 2 metabolites-12-00166-t002:** Biochemical composition of saliva of patients with breast cancer depending on the stage.

Indicator	Group I, *n* = 226	Group II, *n* = 131	Group III, *n* = 75	Group IV, *n* = 55	Kruskal-Wallis Test (H, p)
Protein, g/L	0.68 [0.39; 1.08]	0.56 [0.37; 1.07]	0.58 [0.29; 1.14]	0.84 [0.46; 1.16]	71.46; 0.0000
Urea, mmol/L	10.08 [6.66; 14.12]	9.18 [5.90; 13.10]	9.17 [6.00; 12.08]	8.37 [5.83; 12.30]	71.86; 0.0000
Uric acid, μmol/L	59.6 [23.2; 128.4]	58.3 [26.3; 126.1]	75.6 [16.7; 137.6]	103.8 [39.5; 141.3]	10.87; 0.0280
α-Aminoacids, mmol/L	4.29 [3.90; 4.89]	4.23 [3.91; 4.67]	4.14 [3.86; 4.72]	4.24 [3.84; 4.65]	25.64; 0.0000
NO, μmol/L	29.1 [17.2; 47.5]	30.4 [18.2; 42.5]	22.8 [13.9; 46.0]	32.3 [22.1; 44.6]	18.41; 0.0010
ALP, U/L	76.1 [50.0; 108.7]	68.4 [47.8; 104.3]	71.7 [41.3; 102.1]	82.6 [47.8; 123.9]	16.35; 0.0026
LDH, U/L	1509.0 [829.5; 2088.0]	1452.5 [899.1; 2070.0]	1280.0 [636.8; 2034.0]	1703.0 [972.0; 2279.0]	16.79; 0.0021
Catalase, nkat/mL	3.86 [2.65; 6.25]	3.70 [2.36; 5.22]	3.84 [2.25; 6.11]	3.52 [2.56; 5.51]	11.01; 0.0264
MDA, μmol/L	6.92 [5.73; 8.89]	7.09 [5.90; 9.15]	7.18 [5.94; 9.23]	7.26 [5.81; 8.89]	13.56; 0.0089
GGT, U/L	23.1 [19.8; 26.1]	24.0 [20.4; 27.4]	22.7 [19.9; 26.6]	22.2 [19.4; 26.1]	37.86; 0.0000
Seromucoids, c.u.	0.099 [0.062; 0.163]	0.104 [0.068; 0.165]	0.080 [0.055; 0.135]	0.098 [0.055; 0.138]	8.257; 0.0826
SOD, c.u.	76.3 [34.2; 168.4]	57.9 [28.9; 113.2]	75.0 [36.8; 134.2]	73.7 [39.5; 113.2]	10.31; 0.0355
α-Amylase, U/L	319.0 [157.7; 596.8]	262.2 [124.8; 664.9]	341.5 [111.6; 940.6]	347.8 [115.4; 482.9]	14.84; 0.0050
Ca/P-ratio, c.u.	0.256 [0.185; 0.354]	0.274 [0.196; 0.380]	0.274 [0.221; 0.450]	0.266 [0.185; 0.345]	10.40; 0.0343
SB/TC-ratio, c.u.	0.628 [0.557; 0.698]	0.627 [0.566; 0.696]	0.601 [0.556; 0.683]	0.647 [0.569; 0.765]	9.531; 0.0491

**Table 3 metabolites-12-00166-t003:** Ratios of biochemical indicators of breast cancer patients depending on the stage.

Groups	*n*	UA/Urea, c.u.	UA·CAT/Urea, c.u.
Group I	226	6.47 [1.97; 18.0]	30.4 [6.80; 70.1]
		<0.0001	<0.0001
Group II	131	7.28 [2.15; 17.5]	28.3 [5.50; 65.8]
		<0.0001	<0.0001
Group III	75	10.95 [1.33; 20.2]	32.5 [3.70; 91.6]
		0.0026	0.0015
Group IV	55	11.78 [4.24; 20.3]	42.6 [9.87; 92.2]
		-	0.0416
Control	298	15.05 [4.74; 29.2]	63.3 [17.2; 145.9]

**Table 4 metabolites-12-00166-t004:** Biochemical composition of the saliva of patients with breast cancer, depending on the histological type.

Indicator	Lobular BC, *n* = 86	*p*-Value (Lobular vs. Control)	Ductal BC, *n* = 227	*p*-Value (Ductal vs. Control)	*p*-Value (Ductal vs. Lobular)	Kruskal-Wallis Test (H, p)
Sodium, mmol/L	6.5 [4.1; 11.8]	−22.2% 0.0435	7.7 [4.7; 12.5]	−7.8% 0.1455	0.2577	5.183; 0.0749
Protein, g/L	0.63 [0.36; 1.02]	−20.3% 0.0303	0.69 [0.40; 1.15]	−12.7% 0.1909	0.3301	49.97; 0.0000 *
Urea, mmol/L	9.86 [4.86; 14.33]	+43.7% 0.0001	9.43 [6.03; 12.83]	+37.5% 0.0000	0.4063	41.49; 0.0000 *
Uric acid, μmol/L	56.73 [27.03; 126.92]	−37.2% 0.0219	70.27 [25.00; 140.48]	−22.3% 0.0427	0.4440	4.922; 0.0853
α-Aminoacids, mmol/L	4.16 [3.89; 4.79]	+3.2% 0.0063	4.29 [3.88; 4.89]	+6.5% 0.0000	0.1737	20.75; 0.0000 *
NO, μmol/L	25.4 [15.6; 44.6]	+8.0% 0.3540	26.8 [18.2; 40.7]	+14.0% 0.0355	0.6847	8.614; 0.0135 *
ALP, U/L	69.54 [39.11; 97.79]	+10.3% 0.9190	73.88 [47.81; 108.65]	+17.2% 0.0055	0.0741	12.47; 0.0020 *
MM 254, c.u.	0.206 [0.136; 0.332]	−21.1% 0.0064	0.256 [0.168; 0.382]	−1.9% 0.8093	0.0163	8.728; 0.0127 *
MM 280, c.u.	0.177 [0.118; 0.291]	−19.2% 0.0139	0.200 [0.144; 0.342]	−8.7% 0.8621	0.0267	7.210; 0.0272 *
LDH, U/L	1374.0 [731.8; 2008.0]	+27.7% 0.1590	1532.0 [1022.0; 2217.0]	+42.4% 0.0000	0.0852	20.12; 0.0000 *
Catalase, nkat/mL	3.26 [2.45; 5.49]	−26.6% 0.0013	3.88 [2.52; 6.25]	−12.6% 0.0211	0.2193	7.107; 0.0286 *
TC, c.u.	0.930 [0.822; 1.103]	+5.0% 0.0043	0.904 [0.800; 1.031]	+2.0% 0.0851	0.1197	3.821; 0.1480
SB, c.u.	0.576 [0.490; 0.755]	+7.3% 0.0004	0.555 [0.494; 0.686]	+3.4% 0.0009	0.2518	6.924; 0.0314 *
MDA, μmol/L	6.92 [5.47; 8.29]	+3.7% 0.8404	7.14 [5.90; 9.15]	+7.1% 0.0181	0.1598	10.13; 0.0063 *
GGT, U/L	21.9 [18.3; 24.9]	+5.3% 0.1069	23.4 [19.8; 26.5]	+12.5% 0.0000	0.0424	27.94; 0.0000 *
SOD, c.u.	84.2 [31.6; 152.6]	+45.4% 0.0214	68.4 [39.5; 144.7]	+18.2% 0.0073	0.6601	6.430; 0.0402 *
α-Amylase, U/L	217.4 [113.4; 451.5]	+17.4% 0.3211	304.7 [116.0; 526.4]	+64.5% 0.0077	0.3808	7.360; 0.0252 *
AOA, mmol/L	2.46 [1.29; 3.52]	+4.2% 0.4735	2.23 [1.19; 3.18]	−5.5% 0.0205	0.4471	3.603; 0.1651
Peroxidase, c.u.	0.320 [0.210; 0.750]	−12.3% 0.9743	0.555 [0.290; 0.865]	+52.1% 0.0119	0.1040	4.026; 0.1336
SOD/Catalase-ratio, c.u.	21.5 [9.9; 41.7]	+50.5% 0.0052	19.9 [8.2; 42.8]	+39.3% 0.0010	0.6554	6.739; 0.0344 *
SB/(DC + TC)-ratio, c.u.	0.115 [0.107; 0.153]	+3.6% 0.0006	0.112 [0.103; 0.137]	+0.9% 0.0140	0.1293	4.345; 0.1139
SB/TC-ratio, c.u.	0.644 [0.579; 0.707]	+6.8% 0.0040	0.631 [0.569; 0.723]	+4.6% 0.0006	0.8889	10.32; 0.0057 *

Note. *—the differences between the three groups are statistically significant (Lobular BC, Ductal BC and Control group), *p* < 0.05. AOA—antioxidant activity, SB—Schiff Bases, DC—diene conjugates, TC—triene conjugates, ALP—alkali phosphatase, MDA—malondialdehyde, GGT—gamma glutamyltransferase, LDH—lactate dehydrogenase, MM—middle molecules, SOD—superoxide dismutase.

**Table 5 metabolites-12-00166-t005:** The structure of the study group.

Feature		Breast Cancer, *n* = 487	Control Group, *n* = 298
Age, years		54.5 [47.0; 56.0]	49.3 [43.8; 56.1]
Histological type			
	Ductal	227 (46.6%)	-
	Lobular	86 (17.7%)	-
	Mixed (Ductal + Lobular)	12 (2.5%)	-
	Rare forms	58 (11.9%)	-
	Unknown	104 (21.3%)	-
Clinical Stage			
	Stage I	119 (24.4%)	-
	Stage IIa	123 (25.3%)	-
	Stage IIb	88 (18.1%)	-
	Stage IIIa	55 (11.3%)	-
	Stage IIIb	47 (9.6%)	-
	Stage IV	55 (11.3%)	-
Subtype			
	Luminal A-like	64 (13.1%)	-
	Luminal B-like (HER2+)	230 (47.4%)	-
	Luminal B-like (HER2-)	63 (12.9%)	-
	HER2-positive	38 (7.8%)	-
	Triple-negative	28 (5.7%)	-
	Unknown	64 (13.1%)	-

## Data Availability

The data presented in this study are available on request from the corresponding author. The data are not publicly available because they are required for the preparation of a Ph.D. Thesis.

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
