# Peer review of "Metabolic Features of Saliva in Breast Cancer Patients"

_metabolites, 2022, doi:10.3390/metabo12020166_

Round 1

Reviewer 1 Report

The aim of the work was to study the metabolic characteristics of saliva in breast cancer patients and the subsequent assessment of the potential information content of individual biochemical indicators. The topic of tumor biomarkers is very important, especially by non-invasive methods. Such research can contribute to the development of personal medicine.

This is a large-scale study, but shortcomings must also be pointed out.

 Main disadvantages:

  1. Introduction is too superficial. No previous salivary results have been described.
  2. Discussion need stronger connection with results.  Especially the first part of discussion.
  3. There are described ductal and lobular breast cancer, but the clear analysis of breast cancer subtypes is absent. Really we don`t know nothing about the triple negative breast cancer. The analysis of well known cancer subtypes is needed.
  4. It seems that biomarkers that are technically possible has been determined, so the concept needs to be improved. 

Minor comments:

  1. Abbreviation UA-CAT need special attention.
  2. What are the medium molecular weight toxins.
  3. What is good oral hygiene? Are patients who brush their teeth?

Author Response

The authors are grateful to the reviewers for careful consideration of the manuscript and valuable comments. We hope that thanks to the joint work, the manuscript has become better.

Main disadvantages:

1. Introduction is too superficial. No previous salivary results have been described.

In the above paragraph, all the results were obtained specifically for saliva.

«In the last decade, interest has increased in the use of saliva as an additional test that improves traditional approaches to the medical evaluation of serious systemic diseases, including for the detection and screening of breast cancer [14-17]. Potentially informative salivary tumor markers in breast cancer are epidermal growth factor (EGF), human epidermal growth factor receptor 2 (HER2), vascular endothelial growth factor (VEGF), carcinoembryonic antigen (CEA), CA15-3, and oncogenic tumor suppressor protein (p53) [18-21]. Recent “omics” technologies such as transcriptomics, proteomics, and glycoproteomics allow hundreds of molecules and patterns to be quantified simultaneously to distinguish breast cancer patients from healthy individuals [22-26]. The use of volatile organic compounds in saliva for the diagnosis of breast cancer has been described [27]. It has been reported that hydrophilic metabolites such as amino acids and polyamines can be used to differentiate breast cancer patients from healthy individuals [28-31]. In general, the dynamics of individual components of saliva in breast cancer (trace elements, amino acids, fatty acids, cytokines, etc.) is widely studied [32-35]. However, an integrated approach to the study of saliva for its use in the diagnosis of breast cancer has not yet been implemented.»

We have added the following information, while detailed changes in individual salivary biochemical markers in breast cancer are provided in the Discussion section.

«Saliva is an alternative to blood and other biofluids as a source of potential biomarkers [17]. The main advantage of saliva is its easy non-invasive collection. Studies devoted to the analysis of the proteome and transcriptome of saliva have shown the great potential of this biological fluid for the detection of breast cancer [14, 36]. It has been previously shown that there is a positive correlation between the expression of a number of biomarkers in the blood serum and in saliva, since patients with cancer have a higher expression of specific biomarkers compared to patients without cancer [37-40]. However, most of the studies were conducted at an advanced stage of the disease, when the patient's prognosis is poor, and other diagnostic methods are more effective for making a diagnosis [14]. The identification of cancer at an early stage is extremely important to ensure a good prognosis for the patient and to provide therapy that reduces disease progression and death [41]. This study is one of the largest and includes patients with early stages of breast cancer (226/487). Despite the fact that it would seem that all biomarkers that are technically possible have been identified, their diagnostic significance when determined specifically in saliva has not been evaluated, and combinations of biomarkers and their significance, especially in early breast cancer, have not been considered.»

2. Discussion need stronger connection with results.  Especially the first part of discussion.

We have made appropriate changes to the Discussion section.

3. There are described ductal and lobular breast cancer, but the clear analysis of breast cancer subtypes is absent. Really we don`t know nothing about the triple negative breast cancer. The analysis of well known cancer subtypes is needed.

We deliberately do not touch upon the analysis of changes in the composition of saliva in different molecular biological subtypes of breast cancer, since it is not possible to consider all 5 subtypes within the framework of this manuscript without overloading the manuscript with information. We pointed this out in the limitations section of the study and are now preparing a separate article dedicated to this very issue.

4. It seems that biomarkers that are technically possible has been determined, so the concept needs to be improved. 

We more clearly formulated the concept of our study and made changes to the text of the manuscript.

«This study is one of the largest and includes patients with early stages of breast cancer (226/487). Despite the fact that it would seem that all biomarkers that are technically possible have been identified, their diagnostic significance when determined specifically in saliva has not been evaluated, and combinations of biomarkers and their significance, especially in early breast cancer, have not been considered.»

Minor comments:

1. Abbreviation UA-CAT need special attention.

We have made appropriate changes to the text of the manuscript.

2. What are the medium molecular weight toxins.

MM are substances with their molecular weight ranged from 1500 to 5000 D, which are universal markers of endotoxemia. The nature of MM is inhomogeneous. It has been found that the most chemically identified MM are fragments of endogenous proteins. Perhaps, acyl hydroperoxides and fragments of damaged cell membranes are a primary product to make MM. Being natural biogenic regulators under normal conditions, when in high concentrations, MM have a wide range of pathological effects. MM inhibit mitochondrial respiration, DNA synthesis in alveolar macrophages and lymphocytes, they also inhibit haemoglobin synthesis; they also reduce lactate dehydrogenase activity, etc.

The ultraviolet spectrophotometry of biologic fluids is a basic method to measure MM levels. The wavelength range of 220–255 nm is the peak absorption spectrum of non-aromatic (254 nm) sulphur-containing amino acids (cystine, cysteine, and methionine), 250–256 nm is the maximum of extinctions for purine bases (adenine, guanine). Substances of a catabolic origin, xenobiotics, cellular and tissue debris, and microbial substances are registered in the wavelength range of 238–242 nm. The wavelength range of 238–244 nm is the peak absorption spectrum for urea, uric acid, and creatinine. The degradation product absorption spectrums of albumin, fibrinogen, non-esterified fatty acids, and other substances are close to this maximum. High values of extinctions at the wavelengths of 238, 242, and 246 nm always point out to pathologic processes in the body. At the wavelength of 258 nm, there is the peak absorption of adenosine diphosphate, adenosine monophosphate, adenine, L-valine, L-phenylalanine, while at the wavelength of 280 nm, the light is maximally absorbed by aromatic chromophores, i.e., phenols, tyrosine, tryptophan, and phenylalanine.

In our work, we determine the MM at wavelengths of 254 and 280 nm, they are designated MM 254 and MM 280, respectively.

We have previously published information on medium molecular toxins, so we have added the following information to the text of the manuscript and provided a link «The content of substrates for lipid peroxidation processes (diene conjugates – DC, triene conjugates – TC, Schiff bases – SB, malondialdehyde – MDA) and the level of medium molecular weight toxins (MM) were determined. We determine the MM at wavelengths of 254 and 280 nm, they are designated MM 254 and MM 280, respectively [61].»

3. What is good oral hygiene? Are patients who brush their teeth?

Good oral hygiene means clean teeth free of stuck food; pink gums that don't hurt or bleed when brushing and no bad breath or pain. We took into account that patients did not have untreated dental caries and periodontal disease, which could affect the results of the study.

Patients brushed their teeth and rinsed their mouths with water just before collecting saliva.

Reviewer 2 Report

The manuscript entitled "Exploring Potential Diagnostic and Prognostic Salivary Biomarkers in Cancer" describes the metabolic characteristics of saliva in breast cancer and the subsequent assessment of the potential information content of individual biochemical indicators.

In my opinion, the manuscript is well organized and the results/discussion fundamented with results from literature.

For these reasons, it should be accepted after minor revisions.

Comments:

  • the abbreviations should be described when used for the first time
  • the authors should verify the references once some are not formatted according to the journal
  • Why the values for PCs were so low?
  • Did the authors take into account the pH of saliva from donors?

Author Response

The authors are grateful to the reviewers for careful consideration of the manuscript and valuable comments. We hope that thanks to the joint work, the manuscript has become better.

1. the abbreviations should be described when used for the first time

We have made appropriate changes to the text of the manuscript.

2. the authors should verify the references once some are not formatted according to the journal

We have reviewed the reference list and made changes where necessary.

3. Why the values for PCs were so low?

The PCs values in Figures 2 and 3 are low because there is no statistically significant separation of all groups. The formed groups are large and heterogeneous, so one should not expect an unambiguous division. We consider the results obtained by this method as preliminary for a deeper subsequent analysis in narrow subgroups that are more homogeneous in structure. This is planned to be done in the next phase of the study, as reflected in the limitations.

4. Did the authors take into account the pH of saliva from donors?

We determined the pH of saliva for all patients included in the study, but did not reveal statistically significant differences in this indicator between the groups. In Tables S1-S3, the respective pH values for each group are given and can be consulted.

Round 2

Reviewer 1 Report

The manuscript much is better now, but it still need improving, at preset, the concept needs clarifying in Conclusions or Introduction.

Author Response

The authors are grateful to the reviewers for careful consideration of the manuscript and valuable comments. We hope that thanks to the joint work, the manuscript has become better.

We have made appropriate changes to the text of the manuscript.

Introduction "

The concept of the study is that despite the fact that all biomarkers of breast cancer that are technically possible have been identified, there is no universal diagnostic indicator or a combination of complementary indicators of breast cancer, which is due to the significant heterogeneity of the disease. The search for a combination of potential biomarkers is most promising in saliva, which we have shown earlier using lung cancer as an example. It is especially important to analyze the possibility of detecting early breast cancer using saliva [42].

In the present study, 34 biochemical indicators of saliva in breast cancer were identified and their potential significance was assessed. This study is one of the largest and includes patients with early stages of breast cancer (226/487). The metabolic features of the composition of the saliva of patients depending on the prevalence of the process and the histological type of breast cancer are considered. Biochemical indicators of saliva have been identified, the change of which is typical for the early stages of breast cancer."

Conclusions "It has been shown that the composition of saliva in breast cancer is characterized by certain metabolic features. In accordance with the concept of the study, nine biochemical salivary indicators were identified to differentiate patients with breast cancer from healthy controls with a sensitivity of 91.72%. These indicators are total protein, urea, α-amino acids, nitric oxide, MDA, GGT, alkaline phosphatase, as well as SB/TC-ratio and Na/K-ratio. Biochemical indicators have been identified, the change in which characterizes early breast cancer, which can be used for timely diagnosis in addition to existing methods. For these purposes, the ratios UA/Urea and UA∙CAT/Urea were calculated. It was shown that for ductal breast cancer, the activities of metabolic enzymes of saliva change largely, while for lobular breast cancer, changes in the indicators of enzymatic and non-enzymatic components of antioxidant protection are more pronounced. The obtained data emphasize the expediency of further study of the metabolic characteristics of saliva in relation to the diagnosis of breast cancer."

Round 3

Reviewer 1 Report

The text need a little English editing.

Author Response

The authors are grateful to the reviewers for careful consideration of the manuscript and valuable comments. We hope that thanks to the joint work, the manuscript has become better.

We have made appropriate changes to the text of the manuscript.